# High-Throughput Drug Screening Revealed That Ciclopirox Olamine Can Engender Gastric Cancer Stem-like Cells

**DOI:** 10.3390/cancers15174406

**Published:** 2023-09-03

**Authors:** Diana Pádua, Paula Figueira, Mariana Pinto, André Filipe Maia, Joana Peixoto, Raquel T. Lima, António Pombinho, Carlos Filipe Pereira, Raquel Almeida, Patrícia Mesquita

**Affiliations:** 1i3S—Institute for Research and Innovation in Health, University of Porto, 4200-135 Porto, Portugal; dpadua@ipatimup.pt (D.P.); andre.maia@i3s.up.pt (A.F.M.); jpeixoto@ipatimup.pt (J.P.); rlima@ipatimup.pt (R.T.L.); antonio.pombinho@ibmc.up.pt (A.P.); ralmeida@ipatimup.pt (R.A.); 2IPATIMUP—Institute of Molecular Pathology and Immunology, University of Porto, 4200-465 Porto, Portugal; 3ICBAS—Institute of Biomedical Sciences Abel Salazar, University of Porto, 4050-313 Porto, Portugal; 4IBMC—Institute of Molecular and Cell Biology, University of Porto, 4200-135 Porto, Portugal; 5Pathology Department, Faculty of Medicine, University of Porto, 4200-319 Porto, Portugal; 6CNC—Center for Neuroscience and Cell Biology, University of Coimbra, 3004-517 Coimbra, Portugal; filipe.pereira@med.lu.se; 7Cell Reprogramming in Hematopoiesis and Immunity Laboratory, Molecular Medicine and Gene Therapy, Lund Stem Cell Center, Lund University, BMC A12, 221 84 Lund, Sweden; 8Wallenberg Center for Molecular Medicine, Lund University, 221 84 Lund, Sweden; 9Biology Department, Faculty of Sciences, University of Porto, 4169-007 Porto, Portugal

**Keywords:** gastric cancer, cancer stem cells, ciclopirox, cellular reprogramming, transcription factors, cellular senescence, metabolic reprogramming, cobalt chloride

## Abstract

**Simple Summary:**

Cancer stem cells (CSCs) are thought to be involved in tumor initiation and recurrence, which makes them pivotal therapeutic targets. However, the molecular circuits behind CSC characteristics are not fully understood and the low number of CSCs is an obstacle for conducting tests that explore their properties. Thus, increasing the number of these cells via small molecule-mediated cellular reprogramming seems a valid alternative tool. Using the SORE6-GFP reporter system, based on SOX2/OCT4 activity and incorporated in gastric non-CSCs, we performed a high-throughput drug screen to identify small molecules capable of converting non-CSCs into CSCs. Our results show that ciclopirox olamine (CPX) is able to reprogram gastric cancer cells to a stem-like phenotype via SOX2 expression, reprogram cell metabolism, and induce senescence. Furthermore, these results suggest that the CSC phenotype is a resistance mechanism to CPX treatment, which is being considered as a repurposing drug for cancer treatment.

**Abstract:**

Cancer stem cells (CSCs) are relevant therapeutic targets for cancer treatment. Still, the molecular circuits behind CSC characteristics are not fully understood. The low number of CSCs can sometimes be an obstacle to carrying out assays that explore their properties. Thus, increasing CSC numbers via small molecule-mediated cellular reprogramming appears to be a valid alternative tool. Using the SORE6-GFP reporter system embedded in gastric non-CSCs (SORE6−), we performed a high-throughput image-based drug screen with 1200 small molecules to identify compounds capable of converting SORE6− to SORE6+ (CSCs). Here, we report that the antifungal agent ciclopirox olamine (CPX), a potential candidate for drug repurposing in cancer treatment, is able to reprogram gastric non-CSCs into cancer stem-like cells via activation of SOX2 expression and increased expression of C-MYC, HIF-1α, KLF4, and HMGA1. This reprogramming depends on the CPX concentration and treatment duration. CPX can also induce cellular senescence and the metabolic shift from oxidative phosphorylation (OXPHOS) to glycolysis. We also disclose that the mechanism underlying the cellular reprogramming is similar to that of cobalt chloride (CoCl_2_), a hypoxia-mimetic agent.

## 1. Introduction

Gastric cancer is the fifth most commonly diagnosed cancer and the fourth deadliest cancer in the world [1]. The search for new therapeutic targets has led to the recognition of CSCs as cells responsible for tumor progression, recurrence, and resistance to therapy [2,3,4]. CSCs were first identified in acute myeloid leukemia and have since been identified in most cancers, including gastric cancer [5,6,7]. CSCs constitute a small cell population in the tumor bulk with self-renewal and proliferation capacities and are resistant to various therapies [7,8]. Additionally, studies have shown that these stem-like characteristics are maintained by the adoption of different mechanisms driven by senescence and cellular plasticity [9,10].

Recently, we have transduced two gastric cancer cell lines with the SORE6−GFP reporter system developed by Tang et al., which relies on the activity of the transcription factors (TFs) SOX2 and OCT4 to drive the expression of the GFP reporter, i.e., after incorporation of the reporter system into cells, those that express sufficient levels of SOX2 and/or OCT4 will activate GFP expression, while those that do not express sufficient levels of these TFs remain GFP-negative [7,11]. Two different subpopulations were isolated by fluorescence-activated cell sorting (FACS) and characterized in each gastric cancer cell line: a small subpopulation expressing GFP, SORE6+, which exhibited the expected cancer stem-like features responsible for tumor progression and therapy resistance and can be considered CSCs, in opposition with a GFP-negative subpopulation, SORE6− (non-CSCs). Additionally, SORE6+ cells are enriched in several stemness-related molecules including SOX2, C-MYC, and Notch1, when compared with SORE6− cells [7]. However, the molecular circuitries of CSCs that link stemness with therapy resistance and patient prognosis are not well understood.

Cellular reprogramming is an interesting approach for converting cancer cells into CSCs to produce a greater CSC population for experimental manipulation and to explore the molecular mechanisms that induce the CSC phenotype [12,13]. Reprogrammed CSCs exhibit features and genetic profiles similar to those of CSCs and serve as a useful platform for investigating their origin and molecular functions [13]. Cellular reprogramming of mouse and human somatic cells to a stem-like phenotype has been accomplished by ectopic expression of a cocktail of TFs, such as SOX2, OCT4, KLF4, and C-MYC or SOX2, OCT4, LIN28, and NANOG [14,15,16]. However, the efficiency of cellular reprogramming using TFs remains low because of genetic and epigenetic barriers [14,15]. To overcome this difficulty, small molecules have been tested and used to increase the efficiency of cellular reprogramming and/or to replace some of the main TFs [17,18]. For example, valproic acid, a histone deacetylase inhibitor, increased the efficiency of TF-mediated cellular reprogramming, showing that chromatin modification may be one of the key rate-determining steps during cellular reprogramming [19]. Although most small molecule screenings have been directed towards the discovery of compounds that are capable of killing CSCs, such as salinomycin and monensin [7,20,21], small molecule-mediated cellular reprogramming appears to be a relevant alternative tool for obtaining a larger population of CSCs. It also has appealing advantages including low cost, simplicity, and reversibility [22].

Via high-throughput screening of small molecules, we identified a compound capable of reprogramming gastric cancer cells into CSCs. Here, we report that the small molecule CPX, an iron chelator, has a potent cytotoxic effect on gastric cancer cells, but it can also reprogram gastric non-CSCs to a cancer stem-like cell phenotype, depending on the dose and treatment duration. CPX can promote the metabolic shift from OXPHOS to glycolysis, induce cellular senescence, and increase the expression of several TFs associated with stemness features such as SOX2, C-MYC, HIF-1α, KLF4, and HMGA1. Furthermore, we showed that the mechanism underlying this cellular reprogramming is similar to that of CoCl_2_, a hypoxia-mimetic agent that stabilizes the expression of HIF-1α.

## 2. Materials and Methods

### 2.1. Cell Culture

Four human gastric carcinoma cell lines (AGS SORE6−, AGS SORE6+, Kato III SORE6−, and Kato III SORE6+) established under the scope of previous work [7] were used in this study. Furthermore, the human gastric carcinoma cell lines AGS (ATCC, CRL-1739, Manassas, VA, USA) and Kato III (ATCC, HTB-103, Manassas, VA, USA), from which the SORE6 cell lines were derived, were also used, as well as the human embryonic kidney cell line, HEK293T (kindly provided by Dr. João Relvas from Glial Cell Biology Group, i3S, Porto, Portugal). All gastric cancer cell lines were cultured in RPMI medium 1640 with 25 mM HEPES and GlutaMAX-1 (Gibco, Life Technologies, Thermo Fisher, Waltham, MA, USA) supplemented with 10% fetal bovine serum (FBS) (Biowest, Nuaillé, France). The HEK293T cell line was cultured in DMEM with 4.5 g/L D-glucose, L-glutamine, and pyruvate (Gibco, Life Technologies, Thermo Fisher) supplemented with 10% FBS. All cells were maintained at 37 °C and 5% (*v*/*v*) CO_2_. When the cell cultures reached a confluence of around 80% they were trypsinized with 0.05% Trypsin-EDTA (1X) (Gibco, Life Technologies, Thermo Fisher) and sub-cultured or used in cell-based assays.

### 2.2. High-Throughput Screening of the Prestwick Chemical Library

AGS SORE6− cells [7] were seeded in 384-well plates (CELLSTAR 781091, Greiner, Kremsmünster, Austria) at a density of 5 × 10^4^ cells/mL in RPMI medium 1640 with 25 mM HEPES and GlutaMAX-1 supplemented with 10% FBS. High-throughput screening of 1200 compounds from the Prestwick Chemical Library was performed as described in [7]. Briefly, AGS SORE6− cells were exposed to 1200 small molecules, one compound per well, for 48 h at a final concentration of 8 µM in dimethyl sulfoxide (DMSO) (Applichem, IL, USA). Cells treated with 0.4% DMSO were used as a negative control. The cells were then fixed with 4% paraformaldehyde (PFA) (Alfa Aesar, Haverhill, MA, USA) and stained with Hoechst (1 µg/mL). The INCell Analyzer 2000 (GE Healthcare, Chicago, IL, USA) was used to acquire the images, and image analysis was performed using INCell Investigator software (GE Healthcare). Both image acquisition and analysis were conducted as described in [7]. The image acquisition was performed in an INCell Analyzer 2000 with a Nikon 10×/0.45 NA Plan Fluor objective. Four fields of view were acquired per well covering the whole well. Image analysis was executed using the INCell Investigator software. Briefly, the image analysis workflow consists of the segmentation of the nuclei, from the Hoechst channel, followed by a slight expansion of the nuclear mask to cover a larger area of the cell. The mean pixel intensity of the GFP channel has been extracted for each individual cell. Spotfire software (TIBCO, Palo Alto, CA, USA) was used to visualize the data and perform quality control of the image segmentation and GFP+/GFP− threshold decision. Additionally, a custom MATLAB script (R2018 version, MathWorks, Natick, MA, USA) was generated to automatically obtain specific and conditional data from all the analyzed cells for each condition. Hits were identified using the SORE6−GFP reporter system as a readout for successful cellular reprogramming, indicating that a significant increase in GFP expression in AGS SORE6− cells (GFP-negative) was considered a potential hit. For the final hit validation, the compound was purchased and tested again.

### 2.3. Assessment of Ciclopirox Activity and Cytotoxicity

AGS SORE6− cells were seeded in 96-well plates (CellCarrier-96 Ultra, PerkinElmer, Waltham, MA, USA) at a density of 5 × 10^4^ cells/mL and allowed to attach overnight. Cells were then incubated for 48 h with various concentrations (0.125–16 μM) of ciclopirox olamine (ciclopirox ethanolamine; ciclopirox; CPX) (Abcam 143286, Cambridge, UK) or 0.4% DMSO. To assess cell viability and GFP expression, cells were fixed, stained, and analyzed using the INCell Analyzer 2000 and INCell Investigator software, as described in the Materials and Methods High-throughput screening of the Prestwick chemical library. To further explore the effect of CPX on cell viability and activation of the SORE6−GFP reporter system, AGS SORE6− cells were seeded in 12-well plates at a density of 1 × 10^5^ cells/mL and allowed to settle for 24 h. Then, cells were exposed to 4 or 8 μΜ CPX for 2, 6, 12, 24, or 48 h. For cytotoxicity assessment, culture media was discarded, adherent cells were washed with phosphate-buffered saline (PBS) 1x (grisp, Oporto, Portugal), and 50 μL of PrestoBlue Cell Viability Reagent 1x (ThermoFisher Scientific) prepared in RPMI complete medium was added. Cells were incubated for 1 h and 10 min at 37 °C in a 5% CO_2_ humidified atmosphere, protected from light. The absorbance was measured at 560 nm (using 600 nm as a reference wavelength) using a BioTek Synergy 2 multi-mode microplate reader (Biotek, Winooski, VT, USA). GFP expression was evaluated using a FACSCanto II flow cytometer (BD Bioscience, Franklin Lakes, NJ, USA). The data were analyzed using FlowJo software (version 7.6.1). Cells treated with 8 μM CPX for 24 or 48 h were left in culture for a total of 5 weeks. Every week, cells were collected by trypsinization and centrifuged at 350 rcf for 5 min at room temperature (RT), and cell pellets were used for Western blot and flow cytometry analysis of GFP expression. AGS SORE6− cells and Kato III SORE6− cells were also seeded in 6-well plates at a density of 1.5 × 10^5^ cells/mL and 2 × 10^5^ cells/mL, respectively, and allowed to attach overnight. Cells were then incubated with 2, 4, 8, or 16 μM CPX or 0.4% DMSO for 48 h. After, cells were collected, and cell pellets were used for Western blot and/or real-time PCR analysis. Cells treated with DMSO were used as a negative control. CPX effect was also assessed in CPX-recovered cells. For this, four weeks after exposure to DMSO or 4 or 8 μM CPX, AGS SORE6− and Kato III SORE6− recovered cells were plated in 12-well plates at a density of 1 × 10^5^ cells/mL in RPMI with 10% FBS and allowed to attach overnight. Cells were then exposed for 48 h to 8 μM CPX. For the cytotoxicity assessment, the PrestoBlue assay was performed as described above. The cells were collected and used for flow cytometry and Western blot analyses. Cells treated with DMSO were used as a negative control.

### 2.4. Protein Extraction and Western Blot

The preparation of whole cell extracts and protein quantification were conducted as detailed in [7]. Briefly, 30 μg of total protein extract was separated by standard SDS-PAGE using the Precision Plus Protein Standard Dual Color protein marker (Bio-Rad, Hercules, CA, USA). Subsequently, proteins were transferred to a nitrocellulose membrane (Amersham, GE Healthcare). The membranes were blocked with either 5% non-fat milk or 5% bovine serum albumin (BSA) (Sigma-Aldrich, St. Louis, MO, USA) in tris-buffered saline (TBS)-1% Tween-20 (Sigma-Aldrich) for 1 h at RT. Membranes were incubated overnight at 4 °C with the following primary antibodies: SOX2 (diluted 1:500; Cell Marque, Sigma-Aldrich; AP.CM-371R-15), C-MYC (diluted 1:1000; Cell Signaling Technology, Danvers, MA, USA; (D84C12) #5605), HMGA1 (diluted 1:500; Abcam, Cambridge, UK; ab252930), and β-actin (diluted 1:2000; Santa Cruz Biotechnology, Dallas, TX, USA; sc-47778). After, membranes were washed with TBS-1% Tween-20 and incubated for 1 h at RT with the respective HRP-conjugated secondary antibody: goat anti-rabbit IgG (HRP) (diluted 1:10,000; Cell Signaling Technology, Danvers, MA, USA; #7074) or goat anti-mouse IgG (HRP) (diluted 1:2000; Santa Cruz Biotechnology, Dallas, TX, USA; sc-2005). Signal detection was achieved using the ECL detection kit (Amersham, GE Healthcare). β-actin was used as an internal control.

### 2.5. RNA Extraction and Real-Time PCR

Total RNA extraction using TRI Reagent (Sigma-Aldrich; T9424) was performed according to the manufacturer’s protocol instructions. RNA concentration and quality were evaluated using a NanoDrop ND-1000 spectrometer (V3.5.2 Software). Total RNA (1 μg) was converted to cDNA in a thermocycler (Bio-Rad, Hercules, CA, USA) in a final reaction volume of 20 µL, containing 1 µL of random primers (100 ng/µL) (Invitrogen, Carlsbad, CA, USA), 1 µL of dNTPs (10 mM) (Invitrogen), 2 µL of 10× Reaction Buffer (NZYTech, Lisboa, Portugal), 0.2 µL of RNAseOUT (40 U/µL) (Invitrogen), and 0.5 µL of NZY Reverse Transcriptase (200 U/µL) (NZYTech) in DEPC-treated water (Invitrogen). For the PCR, each reaction was prepared as described in [7]. Primer sequences are shown in Appendix A. The reactions were executed in a 7500 Fast Real-Time PCR System using the software v2.0.6 (Applied Biosystems, Foster City, CA, USA) with the following thermal cycling conditions: 40 cycles of 15 s at 95 °C for denaturation, plus 1 min at 60 °C for annealing and a melting curve program (60–95 °C) with continuous fluorescence measurement. Each reaction was performed in triplicate. Negative controls (without cDNA) were included on each plate and the 18S ribosomal RNA (rRNA) was used as a housekeeping gene. The data were analyzed using the 2^−ΔΔCT^ method [23].

### 2.6. Evaluation of Cobalt Chloride (CoCl_2_) Activity and Cytotoxicity

AGS SORE6− and Kato III SORE6− cells were seeded in 12-well plates at a density of 1 × 10^5^ cells/mL and 1.5 × 10^5^ cells/mL, respectively, and allowed to settle for 24 h. Afterward, cells were incubated with 100, 200, or 300 μM CoCl_2_ for 48 h. Cells were collected and cell pellets were used for Western blot and/or real-time PCR and flow cytometry analysis of GFP expression. Cytotoxicity assessment was performed using PrestoBlue Cell Viability Reagent 1x as described above. Untreated cells (only incubated with complete medium) were used as a negative control.

### 2.7. Clonogenic Assay

AGS SORE6− cells and Kato III SORE6− cells were treated with CPX or CoCl_2_ or were plated in 6-well plates at a density of 300 cells per well and 1500 cells per well, respectively. Cells were left undisturbed for 10 days at 37 °C and 5% (*v*/*v*) CO_2_, except for the fifth day, when the medium was changed. After 10 days, colony formation was visualized by staining the cells with a 1% crystal violet solution diluted in PFA for 30 min. The number of colonies per condition was assessed with the ImageJ software.

### 2.8. Seahorse Real-Time ATP Rate Assay

SORE6− cells were seeded in Agilent Seahorse XF24 24-well cell culture microplates at a density of 6 × 10^4^ cells/mL and incubated with RPMI medium for 24 h. Then, cells were treated with CPX (4 μM and 8 μM) or CoCl_2_ (100 μM, 200 μM, and 300 μM) for 48 h. After, cells were washed and incubated with XF DMEM Medium (DMEM with 5 mM HEPES and without phenol red, sodium bicarbonate, glucose, pyruvate and L-glutamine, pH = 7.4; Agilent Technologies, Santa Clara, CA, USA) supplemented with 10 mM of XF glucose, 2 mM of XF L-glutamine and 1 mM of XF pyruvate, at 37 °C for 1 h in the absence of CO_2_. Next, the real-time ATP rate assay was performed using the Seahorse XF Real-Time ATP Rate Assay Kit (Agilent Technologies) according to Agilent’s recommendations, on the Seahorse XFe24 analyzer (Agilent Technologies). There, the baseline oxygen consumption rate (OCR) was measured, followed by sequential OCR measurements via the sequential injection of Oligomycin and Rotenone/Antimycin A. The ratio between the ATP production rate in the glycolytic pathway (glycoATP) and the ATP production rate from mitochondrial oxidative phosphorylation (mitoATP)—ATP rate index—was used as a parameter to identify changes in metabolic phenotype OCR values. Assays were normalized to the total number of cells in each well and analyzed with Agilent Seahorse Analytics.

### 2.9. Senescence-Associated β-Galactosidase (SA-β-gal) Assay

Cells were plated in 12-well plates at a density of 1 × 10^5^ cells/mL in RPMI with 10% FBS and allowed to attach overnight. Cells were treated with CPX (4 and 8 μM) or CoCl_2_ (100, 200, and 300 μM) for 48 h. After, cells were washed with PBS 1× and fixed with 4% PFA, for 5 min at RT. Subsequently, cells were incubated for 16 h at 37 °C in the absence of CO_2_ in fresh SA-β-gal staining solution prepared as in Dimri et al. [24]. Senescent cells were counted using the ImageJ software (a minimum of 100 cells per condition was considered). Cells treated with DMSO and untreated cells were used as negative controls for CPX and CoCl_2_, respectively.

### 2.10. Cell Proliferation Assay

Cells were cultured as described in [7]. For cell proliferation analysis, the BrdU incorporation protocol was executed as detailed [7]. The percentage of BrdU-positive cells was assessed using a FACSCanto II flow cytometer. The data were analyzed using FlowJo software (version 7.6.1). BrdU incorporation was also observed by immunofluorescence, as in [7]. The coverslips were incubated with the primary antibody BrdU (Bu20a; diluted 1:200; #5292; Cell Signaling Technology) overnight at 4 °C. The goat anti-mouse Alexa Fluor 594 (diluted 1:150; A11032, Thermo Fisher Scientific) was used as the secondary antibody.

### 2.11. Statistical Analysis

The results are expressed as the mean ± standard deviation (SD) from at least three independent experiments. Statistical significance was determined by unpaired two-tailed *t*-test or one-way ANOVA followed by Tukey’s post hoc test using the GraphPad Prism 5.0 software (GraphPad Software, La Jolla, CA, USA). Differences were considered significant at *p* ≤ 0.05.

## 3. Results

### 3.1. CPX Activates the SORE6−GFP Reporter System Embedded in Gastric Non-CSCs

To identify new drugs that could be used to reprogram gastric cancer cells into CSCs, we screened 1200 compounds from the Prestwick chemical library in previously obtained AGS SORE6− cells (non-CSCs) [7] and evaluated whether any compounds could activate the SORE6−GFP reporter system incorporated into these cells (Figure 1A). Since AGS SORE6− cells are GFP-negative, contrary to AGS SORE6+ cells [7], compounds that induced GFP expression were considered hits. Of the 1200 compounds, five appeared to activate GFP expression in AGS SORE6− cells (Figure 1B). Further analysis showed that merbromin (mercurochrome) was a false positive, as it emitted yellow-green fluorescence when excited with blue or violet-blue light [25]. Rebamipide, vorinostat, and topotecan, although inducing over 10% GFP expression, were shown to strongly decrease cell viability (over 98%) (Figure 1B,C). Therefore, from the compounds evaluated, only the small molecule CPX appeared to be a strong candidate for cellular reprogramming. This compound activated the SORE6−GFP reporter system in more than 10% of cells while decreasing cell viability by approximately 89% (Figure 1B–D). To further confirm that CPX was a potential cellular reprogramming agent, dose–response assays were performed to assess cytotoxic effects and induction of GFP-positive cells. After 48 h of treatment, CPX caused a dose-dependent cell viability decrease (Figure 1E). Moreover, increasing CPX concentrations of 4, 8, and 16 μM induced GFP expression in 3%, 7%, and 11% of the surviving cells, respectively (Figure 1F). To rule out the possibility that the observed GFP expression after CPX treatment was related to its autofluorescence, we incubated AGS wt cells with 8 μM of CPX for 48 h and confirmed there was no fluorescence increase (Appendix A). Importantly, we determined that CPX treatment induced the expression of SOX2 in AGS SORE6− cells, which led to the activation of the SORE6−GFP reporter system (Figure 1G). These results are the first indication that CPX plays a promising role in gastric CSC reprogramming via activation of SOX2 expression. Although the highest concentration of CPX (16 µM) resulted in a higher percentage of GFP-positive cells, its cytotoxic effect was high and therefore this concentration was not evaluated in further experiments.

### 3.2. Activation of the SORE6-GFP Reporter System via CPX Seems to Be Strongly Associated with Increased SOX2 Expression

To further explore the potential of CPX for cellular reprogramming, we performed a time-course analysis (2, 6, 12, 24, and 48 h) of AGS SORE6− cells treated with 4 and 8 µM CPX. Cell evaluation at 2 and 6 h treatment, showed no effect of CPX in the activation of the reporter system nor on cell viability (Figure 2A,B). At 12 h treatment, only 8 µM CPX significantly activated the reporter system (Figure 2A), while at 24 h, treatment with both 4 and 8 µM CPX induced a significant increase in the percentage of GFP-positive cells (approximately 1% and 4%, respectively). The maximum effect was observed, after 48 h treatment, for both concentrations, being approximately 3% for 4 µM and 8% for 8 µM (Figure 2A). Nevertheless, the CPX effect on cellular viability significantly increased over time at both concentrations (Figure 2B). Next, we evaluated how long the SORE6−GFP reporter system remained active after exposure to CPX. In AGS SORE6− cells treated with 8 μM for 24 h, GFP expression was maintained for approximately one week, increasing to approximately 3 weeks when cells were treated with CPX for 48 h (Figure 2C). Furthermore, our results suggest an association between the decreased GFP and SOX2 expression (Figure 2D). Thus, activation of the SORE6−GFP reporter system via CPX seems to be strongly associated with the increased SOX2 expression. Since GFP expression lasted longer with the 48 h treatment, this was the exposure time chosen for our further analysis, which included another non-CSC gastric cell line (Kato III SORE6−). Similar to AGS SORE6− cells, Kato III SORE6− cells treatment with CPX (specifically with 8 µM) significantly activated the reporter system in approximately 4% of the cells (Figure 2E,F). A dose-dependent increase in SOX2 and C-MYC expression was evident in both cell lines (Figure 2G). Although CPX significantly decreased the viability of Kato III SORE6− cells, these cells appeared to be more resistant to CPX treatment than AGS SORE6− cells (Figure 2H). This difference between these two cell lines was also observed in the colony-formation assay (Figure 2I).

### 3.3. CPX Promotes a Metabolic Shift from Oxidative Phosphorylation (OXPHOS) to Glycolysis and Induces Cellular Senescence

CSCs show metabolic plasticity, and therefore, one of the relevant characteristics of CSCs is their metabolic phenotype [26]. To assess the effect of CPX on AGS SORE6− cellular metabolism and ATP production rate we used the Seahorse Real-Time ATP rate assay. Briefly, Oligomycin inhibits mitochondrial ATP synthesis by decreasing the rate of oxygen consumption (OCR), which allows quantification of the rate of mitochondrial ATP production. Next, Rotenone/Antimycin A led to complete inhibition of mitochondrial respiration, allowing the assessment of the extracellular acidification rate (ECAR), which allows the determination of glycolytic ATP production. Our analysis showed that 48 h of CPX treatment (4 or 8 μM) decreased OCR and increased ECAR (Figure 3A,B). CPX treatment significantly decreased the rate of mitochondrial OXPHOS ATP production at both concentrations compared to the control (Figure 3C), whereas the rate of glycolytic ATP production was significantly increased in cells treated with 8 μM CPX for 48 h (Figure 3C). The ratio of mitochondrial to glycolytic ATP production was calculated and found to be significantly decreased in cells treated with both CPX concentrations (Figure 3D). Additionally, the percentage of glycolysis also increased significantly in the CPX-treated cells (Figure 3E). CPX has been described as an inducer of cellular senescence [27,28]. Senescence has been linked to the promotion of cancer cell reprogramming towards stemness [9]. It can be induced by unrepaired and improper expression of oncogenes, DNA damage, or other cellular stresses [29]. Furthermore, despite the loss of their proliferative capacity, senescent cells have high metabolic activity, preferring energy production from glycolysis rather than OXPHOS [30]. Therefore, we investigated whether CPX triggered senescence in SORE6− cells. Senescence induction was assessed using β-galactosidase (SA-β-gal) activity, a well-established senescence marker [24]. Exposure to 8 µM CPX for 48 h caused a significant increase in SA-β-gal-labeled cells in both AGS SORE6− (approximately 30%) and Kato III SORE6− cells (around 25%) (Figure 3F). Additionally, CPX treatment also led to an apparent increase in cell size. AGS SORE6− cells treated with 4 and 8 μM CPX appeared to be larger than control cells and the same was observed in Kato III SORE6− cells treated with 8 μM CPX (Figure 3F). Furthermore, results from BrdU incorporation by flow cytometry showed that CPX treatment significantly reduced proliferation in both cell lines (Figure 3G).

### 3.4. CPX-Induced Cellular Senescence Is Reversible and Recovered Cells Become More Resistant to CPX and More Prone to SORE6−GFP Reporter System Re-Activation

As mentioned above, senescence drives the reprogramming of cancer cells and promotes CSC generation [9]. Recent studies have suggested that this is achieved by the ability of senescent cells to re-enter the cell cycle to become CSCs, reversing the “irreversible” arrest of cell division and proliferation [31]. Thus, we investigated whether CPX-treated cells could overcome senescence and acquire some characteristics of CSCs, such as increased proliferative activity and drug resistance. AGS SORE6− cells and Kato III SORE6− cells treated for 48 h with 4 or 8 µM CPX were cultured for 4 weeks. SA-β-gal activity was then evaluated with no significant differences being observed in the percentage of SA-β-gal positive cells when comparing cells recovered from exposure to DMSO or 4 or 8 µM CPX (Figure 4A and Appendix A). Interestingly, while evaluation of cells immediately after 48 h treatment with 8 µM CPX, showed 30% of AGS SORE6− cells and 25% of Kato III SORE6− cells as positive for SA-β-gal, 4 weeks later, only 11% and 7%, respectively, were positive for SA-β-gal, thus revealing a possible reversal of the senescent state (Figure 3F, Figure 4A and Appendix A). AGS SORE6− cells recovered from CPX treatment also regained the ability to proliferate; specifically, cells recovered from 8 µM CPX treatment showed significantly improved proliferation activity compared with those recovered from DMSO exposure (negative control) (Figure 4B and Appendix A). We also re-exposed AGS SORE6− and Kato III SORE6− recovered cells to 8 µM CPX for 48 h to verify the impact of a new CPX treatment on the activation of the SORE6−GFP reporter system and cell viability. Remarkably, the new treatment with CPX greatly increased the activation of the reporter system. In AGS SORE6− recovered cells, previously treated with 4 µM or 8 µM, the re-exposure to 8 µM CPX for 48 h resulted in 28% and 35% GFP-positive cells, respectively (Figure 4C). In addition, re-exposure to CPX revealed that cells recovered from previous CPX treatment acquired resistance to the compound (Figure 4D). Similar results were obtained in Kato III SORE6− recovered cells (Appendix A). Based on the SORE6−GFP reporter system activation, our results suggest that cells treated with CPX acquire plasticity by dynamically switching between SORE6+ (cancer stem-like cells) and SORE6− (non-CSCs) phenotypes.

### 3.5. Activation of the Reporter System SORE6-GFP by CPX Has a Mechanism Similar to That of the Hypoxia Mimetic CoCl_2_

CPX has been described as a HIF-1α stabilizer and SOX2 has been reported to interact with the HIF-1α promoter (23, 24). Our results confirmed that treating AGS SORE6− cells for 48 h with 8 µM CPX significantly increased the expression of HIF-1α (Figure 5A). Since HIF-1α expression is higher in CSCs and the AGS SORE6+ cells expressed significantly more HIF-1α than AGS SORE6− cells (Figure 5A), we decided to explore whether this mechanism could be responsible for inducing the expression of SOX2 and consequently activating the SORE6−GFP reporter system. We started by stabilizing/increasing HIF-1α expression using CoCl_2_, a hypoxia-mimetic agent capable of increasing the expression of stem cell markers (25, 26). To achieve this, we treated AGS SORE6− and Kato III SORE6− cells with different CoCl_2_ concentrations for 48 h. Activation of the SORE6-GFP system was observed in both cell lines after treatment with 200 and 300 µM CoCl_2_ (approximately 21% and 19%, respectively, in AGS SORE6− cells, and 2% and 7%, respectively, in Kato III SORE6− cells) (Figure 5B,C). In addition, an increase in SOX2, C-MYC, and HIF-1α expression was observed in both cell lines (Figure 5D,E). Although not as cytotoxic as CPX, CoCl_2_ had dose-dependent cytotoxicity in AGS SORE6− cells, whereas Kato III SORE6− cells were more resistant to treatment (Figure 5F). To rule out the possibility that the observed GFP expression after CoCl_2_ treatment was autofluorescence, we incubated AGS wt cells with 200 µM CoCl_2_ for 48 h, and no fluorescence was observed (Appendix A). Additionally, cells treated with 100, 200, and 300 µM CoCl_2_ showed CPX-like changes in OCR, ECAR, ATP production and rate, and percentage of glycolysis (Appendix A–E). CoCl_2_ treatment significantly decreased the rate of mitochondrial OXPHOS ATP production as well as the ratio of mitochondrial to glycolytic ATP production (Appendix A). The rate of glycolytic ATP production increased in cells treated with 200 µM CoCl_2_ for 48 h (Appendix A). Overall, our results showed similar effects of CPX and CoCl_2_. Furthermore, CoCl_2_ also induced cellular senescence with consequent loss in cell proliferation (Appendix A). Because CPX and CoCl_2_ increased the expression of SOX2, C-MYC, and HIF-1α (Figure 2G and Figure 5A,D,E), we explored whether these compounds also increased the expression of other relevant stemness markers in SORE6+ and SORE6− cells. Real-time PCR results revealed that GATA6, RELA, STAT3, KLF4, OCT1, and HMGA1 were significantly upregulated in AGS SORE6+ cells. In particular, HMGA1 was expressed more than 2-fold in AGS SORE6+ cells than in AGS SORE6− cells (Appendix A). In Kato III SORE6+ cells, C-MYC, HMGA1, and KLF4 were significantly upregulated. Specifically, KLF4 was expressed more than 6-fold in Kato III SORE6+ cells than in Kato III SORE6− cells (Appendix A). Notably, AGS SORE6− and Kato III SORE6− cells treatment with 8 µM CPX (for 48 h) also significantly increased the expression of HMGA1, C-MYC, and HIF-1α (Figure 5G). In AGS SORE6− cells, CPX treatment also increased KLF4 expression (Figure 5G). Similarly, 200 µM CoCl_2_ treatment significantly increased C-MYC, HIF-1α, HMGA1, KLF4, and NANOG expression in AGS SORE6− cells (Figure 5G). In Kato III SORE6− cells, 300 µM CoCl_2_ upregulated C-MYC, HIF-1α, KLF4, NANOG, and OCT4 (Figure 5G).

## 4. Discussion

CSCs constitute a very small population of cells present in the tumor. Evidence from experimental models and clinical studies indicates that CSCs are responsible for therapy resistance and disease recurrence [32,33,34]. Therefore, studying their characteristics is essential to understand how to target them. Since this is a very small population, several strategies have been considered to increase cell number in order to perform functional assays. One of these strategies is cellular reprogramming, which refers to the process of reverting specialized cells into induced pluripotent stem cells by rewiring transcriptional and epigenetic networks. During cellular reprogramming, the cell behavior, DNA methylation, histone modifications, and gene expression profiles can undergo several modifications [35,36,37]. In cancer, cellular reprogramming is an interesting approach to convert malignant cells into benign cells, but also to reprogram cancer cells into CSCs in order to generate a greater CSC population to evaluate their biological properties to improve cancer diagnosis and treatment [12,13,38].

Here, we used our previously established and characterized CSC model [7], which is based on SOX2 and OCT4 activity (SORE6+ (CSCs) vs. SORE6− (non-CSCs)), as a robust platform for the screening of small molecules that can impose CSC properties on SORE6− cells. Activation of the SORE6−GFP reporter system was used as a readout for successful cellular reprogramming. Our screening approach identified CPX, a broad-spectrum synthetic antifungal agent with antibacterial and anti-inflammatory activities [39], as an activator of the reporter system in SORE6− cells. Since SORE6− cells do not become GFP+ over time when they are treated with DMSO or medium, we are confident that CPX is inducing cellular reprogramming. Furthermore, we confirmed that this activation was associated with increased SOX2 expression. CPX also displayed a potent cytotoxic effect on the tested gastric cancer cells, decreasing cell viability and colony formation. Since CPX has been shown to have anti-tumorigenic properties and repress cell growth, migration, and invasion [40,41,42,43,44,45], its repurposing for cancer treatment has been already proposed [46]. Clinical studies have been performed on patients to assess CPX efficiency in the cancer setting [46,47]. However, despite the anti-tumor effect of CPX, we also observed that cells can easily acquire resistance to CPX treatment, which can be problematic for the use of this compound as a chemotherapeutic agent.

CPX has been reported to be associated with OXPHOS inhibition, and it also showed pro-senescent activity, linked to its activity as an iron (Fe^3+^ and Fe^2+^) chelator [27]. Our results fully corroborate the ability of CPX to decrease OXPHOS and induce cellular senescence. Although this induction of cellular senescence may represent a blockage in tumor growth, it may also have the opposite effect of increasing tumor progression and resistance to therapy. This could be related to the close link between cellular senescence and the emergence of CSCs [48]. Accordingly, our results support the idea that senescence can be overcome, and cells can regain proliferative activity. Senescence can be induced by chemotherapeutic agents (e.g., cisplatin and doxorubicin) and radiotherapy [49,50]. Senescent cells acquire a senescence-associated secretory phenotype that can lead to the senescence-associated reprogramming of non-CSCs towards a stem-like phenotype [9,51,52]. It is still in debate if the senescent cells are the ones that reprogram or if they create an environment prone to cellular reprogramming via the secretion of factors that induce reprogramming in neighboring cells, which are known as Senescence Associated Secretory Proteins (SASPs). Recently, Lu and Zhang [49] reported that radiochemotherapy triggers DNA damage response in gastric cancer cells reprogramming them into gastric cancer stem-like cells via the EID3-NAMPT-Wnt/β-catenin axis. They also showed that treatment with doxorubicin or cisplatin increased the expression of stemness markers ALDH1, OCT3/4, SOX2, C-MYC, and KLF4 in gastric non-CSCs [49]. Our results also indicated that treatment of gastric non-CSCs with CPX upregulated the expression of several stemness markers, including SOX2, C-MYC, KLF4, HMGA1, and HIF-1α. Moreover, we postulate that the mechanism of cellular reprogramming exhibited by CPX may be similar to that of CoCl_2_, a well-known hypoxia-mimetic agent that stabilizes HIF-1α expression by blocking its degradation [53]. CoCl_2_ treatment promotes the expression of stem cell markers such as SOX2, OCT4, C-MYC, and NANOG in a dose-dependent manner [54,55]. Our results confirmed that CoCl_2_ treatment increased the expression of SOX2, C-MYC, KLF4, OCT4, NANOG, and HMGA1, showing similarities with the effect of CPX in promoting CSCs by upregulating the same set of TFs. Lopez-Sánchez et al. reported that treatment of colon cancer cells with CoCl_2_ gave rise to polyploid giant cells (larger cells and greater nuclear size) with cancer stem-like features [56]. Interestingly, we also observed an increase in cell size after CPX and CoCl_2_ treatments in the two gastric cancer cell lines we have studied. Similarly to CPX, CoCl_2_ treatment also demonstrated pro-senescent activity and promoted a metabolic shift towards glycolysis, reducing OXPHOS. Metabolically, CSCs can change the way they obtain ATP, depending on the type of tumor and the conditions of the tumor microenvironment. For example, in hypoxic environments, CSCs appear to prefer producing ATP via glycolysis rather than OXPHOS, even in the presence of oxygen (Warburg effect) [57,58,59]. With hypoxia being a common feature of the microenvironment of solid tumors, it is worth mentioning that hypoxia and its master regulators—hypoxia-inducible factors (HIFs)—participate in stem-like maintenance in various cancers and dysregulate stemness genes such as SOX2 [60,61].

## 5. Conclusions

Together, our results, although supporting the powerful cytotoxic effect of CPX on tumor cells, also recommend care if its therapeutic use is intended, since it may kill most tumor cells but may also convert surviving cells into CSCs. At least at certain concentrations and exposure times, the potential for cellular reprogramming is present via induction of cellular senescence (which can be reversed), metabolic plasticity, and increased expression of stemness markers (e.g., SOX2, C-MYC, KLF4, HMGA1). Therefore, CPX may be a useful tool for obtaining gastric cancer stem-like cells from non-CSCs, particularly when cells previously treated with CPX are re-exposed to the compound. On the other hand, in an anti-tumor context, to potentiate its cytotoxic effect and avoid cellular reprogramming of gastric non-CSCs towards a stem-like state, adjustments in CPX concentration and exposure period should be evaluated.

## Figures and Tables

**Figure 1 cancers-15-04406-f001:**
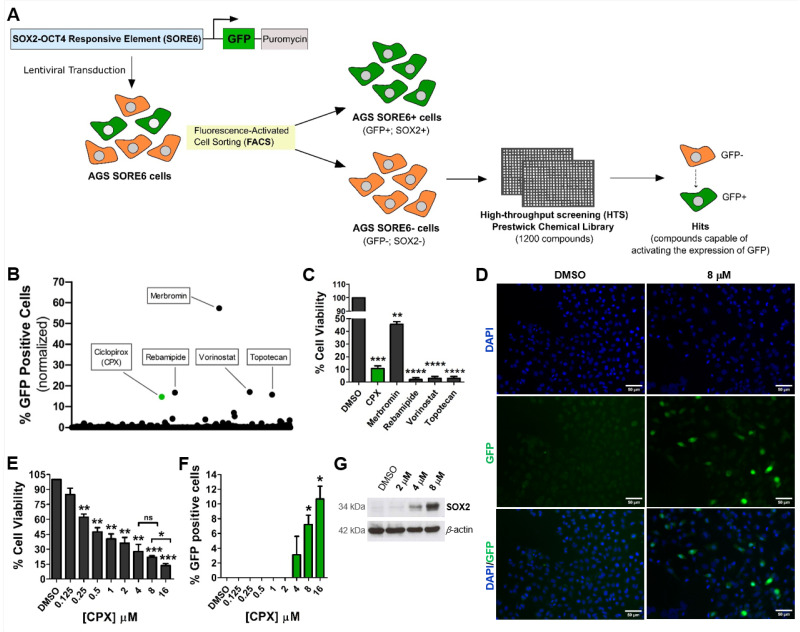
CPX is able to activate the SORE6−GFP reporter system incorporated in the AGS SORE6− cells by inducing SOX2 expression. (**A**) Scheme of the general approach used to identify compounds capable of transforming SORE6− cells (GFP-negative, non-CSCs) into SORE6+ cells (GFP-positive, CSCs). (**B**) High-throughput screening results showing the percentage of GFP-positive cells. A total of 1200 small molecules from the Prestwick chemical library were screened. (**C**) AGS SORE6− cell viability results after 48 h of treatment with CPX, merbromin, rebamipide, vorinostat, and topotecan. (**D**) Representative immunofluorescence images of reporter system activation after treatment with 8 µM CPX for 48 h (cells become GFP-positive cells), scale bar = 50 µm. (**E**) Percentage of cell viability and (**F**) percentage of GFP-positive AGS SORE6− cells after incubation for 48 h with increasing CPX concentrations (0.125–16 µM). The percentage of GFP-positive cells was calculated from viable cells. (**G**) SOX2 expression in AGS SORE6− cells treated with 2, 4, or 8 µM CPX for 48 h, evaluated by Western blot, β-actin was used as an internal control, pictures show cropped areas of Western blots, the whole images are included in the Appendix A. DMSO treatment was used as the negative control (CPX solvent/vehicle). Results are mean ± SD of at least three independent experiments. ns: not significant; * *p* ≤ 0.05; ** *p* ≤ 0.01; *** *p* ≤ 0.001; **** *p* ≤ 0.0001.

**Figure 2 cancers-15-04406-f002:**
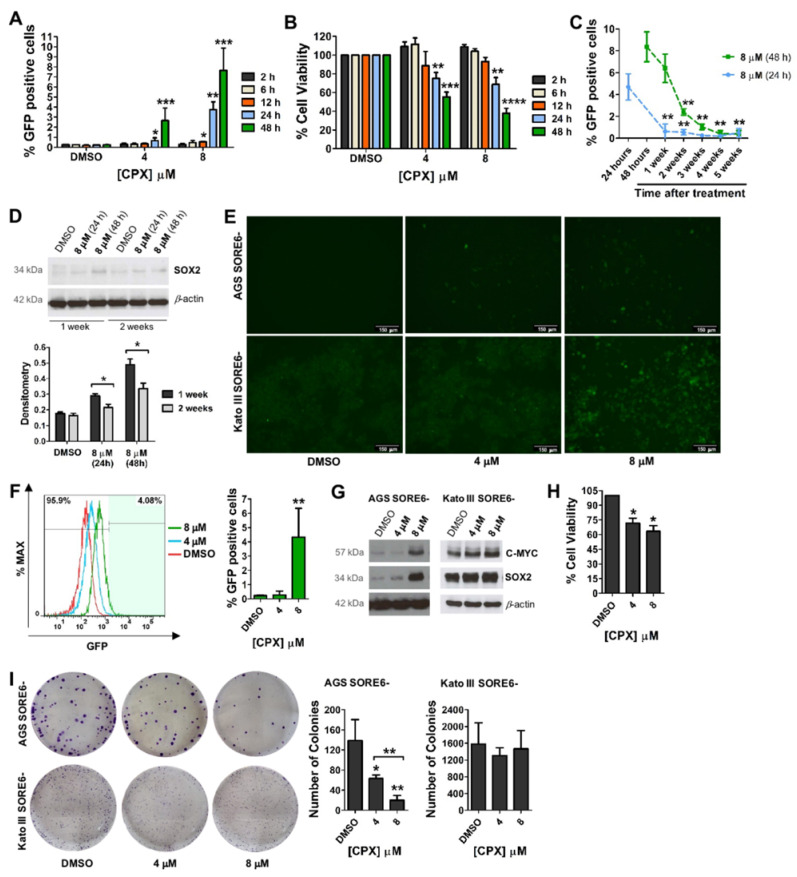
SORE6−GFP reporter system activation by CPX is not permanent and is associated with the consequent loss of SOX2 expression. (**A**) Percentage of GFP-positive cells and (**B**) cell viability after treatment of AGS SORE6− cells with 4 or 8 µM CPX for 2, 6, 12, 24, and 48 h. (**C**) Percentage of GFP-positive cells evaluated one to five weeks after treatment of AGS SORE6− cells with 8 µM CPX for 24 or 48 h. (**D**) SOX2 expression in AGS SORE6− cells after one and two weeks of treatment with 8 µM CPX for 24 or 48 h, evaluated by Western blot (and respective densitometry analysis (intensity ratio: “gene of interest”/β-actin)). (**E**) Representative fluorescence live cell images of SORE6−GFP reporter system activation in AGS SORE6− and Kato III SORE6− cells after treatment with 4 or 8 µM CPX for 48 h, scale bar = 150 µm. (**F**) Percentage of GFP-positive cells after treatment of Kato III SORE6− cells with 4 or 8 µM CPX for 48 h, following the respective flow cytometry analysis. (**G**) SOX2 and C-MYC expression in AGS SORE6− and Kato III SORE6− cells treated with 4 or 8 µM CPX for 48 h, evaluated by Western blot. (**H**) Cell viability after treatment of Kato III SORE6− cells with 4 or 8 µM CPX for 48 h. (**I**) Colony formation ability of AGS SORE6− and Kato III SORE6− cells after treatment with 4 or 8 µM CPX for 48 h, representative images of the colonies are shown. β-actin was used as a loading control. Pictures show cropped areas of Western blots, the whole images are included in the Appendix A. DMSO was used as the negative control. Results are mean ± SD of at least three independent experiments. * *p* ≤ 0.05; ** *p* ≤ 0.01; *** *p* ≤ 0.001; **** *p* ≤ 0.0001.

**Figure 3 cancers-15-04406-f003:**
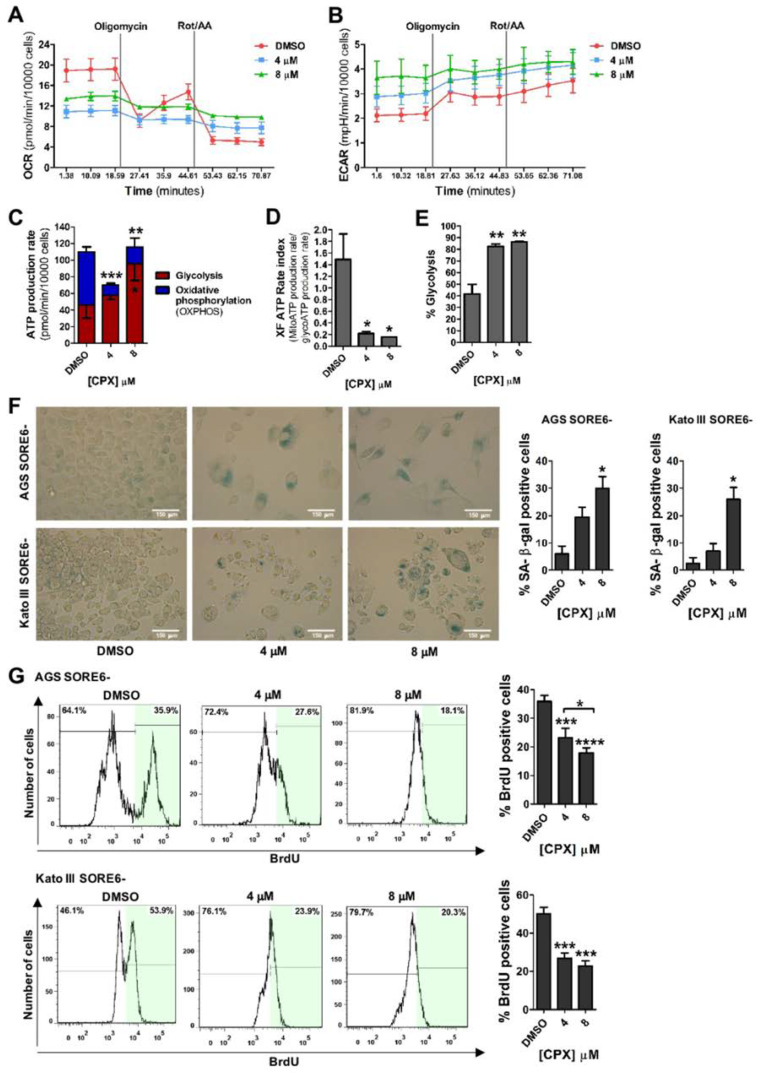
CPX treatment inhibits ATP production by OXPHOS and induces cellular senescence with the consequent loss of proliferative activity. (**A**) Representative real-time measurements of the kinetic profile of the oxygen consumption rate (OCR), (**B**) representative real-time measurements of the kinetic profile of the Extracellular Acidification Rate (ECAR), (**C**) metabolic flux analysis showing quantification of mitochondrial ATP production and glycolytic ATP production, (**D**) ATP production rate index calculated from the data showed in C (i.e., mitochondrial ATP production rate/glycolytic ATP production rate), and (**E**) percentage of glycolysis following treatment with 4 or 8 μM CPX for 48 h. (**F**) Percentage of SA-β-gal-positive cells in AGS SORE6− and Kato III SORE6− cells treated with 4 or 8 µM CPX for 48 h, following representative images of SA-β-gal marking. Scale bar = 150 μm. (**G**) Percentage of BrdU positive cells in AGS SORE6− and Kato III SORE6− cells treated with 4 or 8 µM CPX for 48 h, following representative flow cytometry analysis. DMSO was used as a negative control. Results are mean ± SD of at least three independent experiments. * *p* ≤ 0.05; ** *p* ≤ 0.01; *** *p* ≤ 0.001; **** *p* ≤ 0.0001.

**Figure 4 cancers-15-04406-f004:**
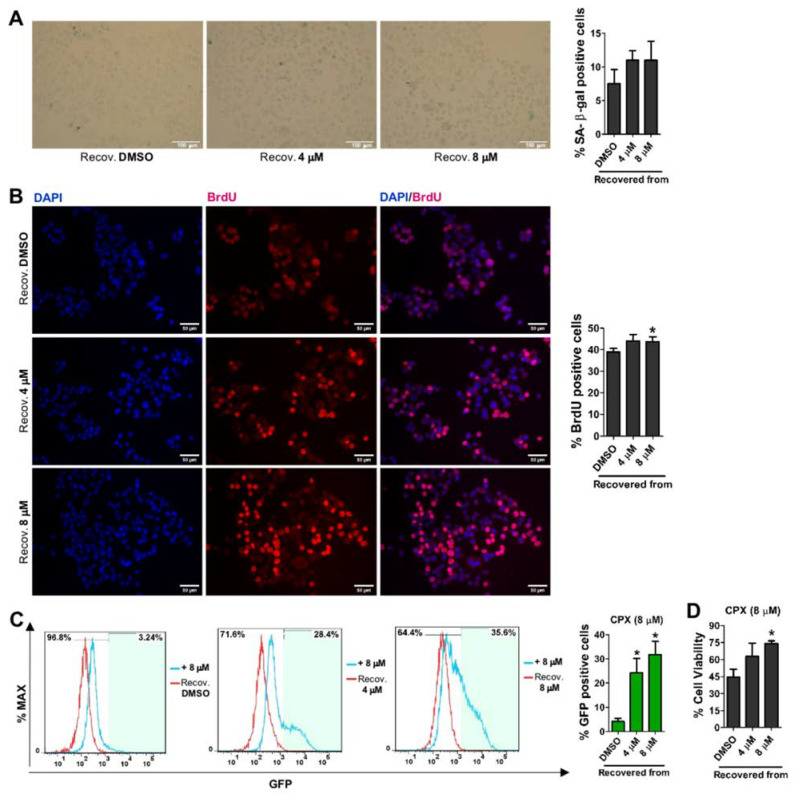
CPX-induced cellular senescence is reversible, and after a new treatment, activation of the SORE6−GFP reporter system and resistance to treatment increase. (**A**) Percentage of SA-β-gal positive cells in AGS SORE6− cells 4 weeks after recovery from treatment with DMSO, 4 or 8 µM CPX for 48 h, following the representative images of SA-β-gal marking. Scale bar = 150 µm. (**B**) Percentage of BrdU positive cells, evaluated by flow cytometry, in AGS SORE6− recovered cells, following representative immunofluorescence images displaying the BrdU incorporation. Scale bar = 50 µm. (**C**) Percentage of GFP-positive cells, following the respective flow cytometry analysis, and (**D**) cell viability in AGS SORE6− recovered cells after new treatment with 8 µM CPX for 48 h. DMSO was used as the negative control. Results are mean ± SD of at least three independent experiments. * *p* ≤ 0.05.

**Figure 5 cancers-15-04406-f005:**
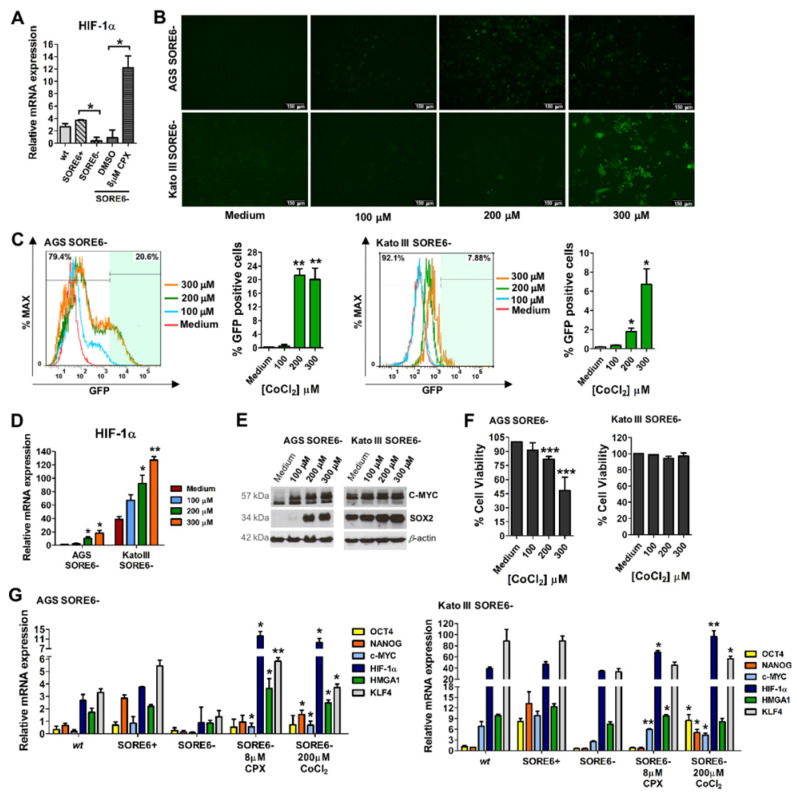
CPX and CoCl_2_ treatment can effectively activate the SORE6−GFP reporter system and increase the expression of several transcription factors associated with stemness. (**A**) Real-time PCR analysis exhibiting the mRNA levels of HIF-1α in AGS wt, AGS SORE6+, AGS SORE6− and AGS SORE6− cells after 48 h treatment with DMSO or 8 µM CPX. (**B**) Representative fluorescence images of SORE6−GFP reporter system activation, scale bar = 150 µm, and (**C**) percentage of GFP-positive cells in AGS SORE6− and Kato III SORE6− cells treated with 100, 200, or 300 µM CoCl_2_ for 48 h, following the respective flow cytometry analysis. (**D**) Real-time PCR analysis exhibiting the mRNA levels of HIF-1α in AGS SORE6− and Kato III SORE6− cells after treatment with 100, 200, or 300 µM CoCl_2_ for 48 h. (**E**) SOX2 and C-MYC expression, evaluated by Western blot and (**F**) a percentage of cell viability after treatment of AGS SORE6− and Kato III SORE6− cells with 100, 200, or 300 µM CoCl_2_ for 48 h; pictures show cropped areas of Western blots, the whole images are included in the Appendix A (**G**) Real-time PCR analysis of mRNA levels of the transcription factors OCT4, NANOG, C-MYC, HIF-1α, HMGA1, and KLF4 in wt, SORE6+, SORE6−, and SORE6− cells treated with 8 µM CPX or 200 µM CoCl_2_ for 48 h. DMSO or medium was used as a negative control. wt corresponds to the parental cell line (AGS or Kato III). Real-time PCR results were normalized to 18S expression. β-actin was used as a loading control. Results are mean ± SD of at least three independent experiments. * *p* ≤ 0.05; ** *p* ≤ 0.01; *** *p* ≤ 0.001.

## Data Availability

The data obtained and analyzed during this work are presented in the published article together with its Appendix A. The data are also available from the authors upon request.

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
