# Peer review of "High-Throughput Drug Screening Revealed That Ciclopirox Olamine Can Engender Gastric Cancer Stem-like Cells"

_cancers, 2023, doi:10.3390/cancers15174406_

Round 1

Reviewer 1 Report

The current manuscript presents original research work by Pádua and co-authors which describes the employment of small molecule ciclopirox olamine (CPX) to convert non-stem gastric cancer cells into cancer stem-like cells CSCs using a SOX2/Oct4/GFP reporter system. The authors claim that such reprogramming enriches the sub-population of CSC within the pool of gastric cancer cells and further switches cell metabolism from oxidative phosphorylation to glycolytic ATP production and initiates cellular senescence. Furthermore, the authors show that the CSC phenotype allows gastric cancer cells to withstand CPX treatment.

The manuscript is well-written, relevant to the field and the journal’s scope, and if published the approach will be an asset to the research community. However, several points need to be either verbally clarified or experimentally validated before the article could be considered for publication.

Specific comments:

1)      The authors cite previously reported SORE6-GFP reporter system (citation 7, Tang et al.) in their Introduction, line 56. I believe it is essential to provide more explanation on the system in the intro of the current manuscript in order for the readers to understand what AGC SORE6- and SORE6+ cells stand for. I read both the current manuscript AND the article by Tang et al. and it is still not clear to me specifically what the SORE6- cells are. Are these the successfully transduced cells (aka cells that do have the reporter system successfully incorporated) but the SOX2 and OCT4 are not activated there resulting in no GFP signal? In that case, is SOX2/OCT4 expression inducible in this system and how is it induced? Alternatively, are these the cells that stayed non-transduced (did not incorporate the reporter system after the transduction procedure)? In this scenario, how can they start expressing the GFP signal following small molecule induction if the SORE6-GFP reporter system is not there, to begin with? This is the most essential comment to address as the whole manuscript is based on the understanding that the cells either express SOX2/OCT$ or they do not.

Related to that, Figure 1A requires an update. The current Figure 1A is read as if the bulk of AGS cells was subjected to transduction with SORE system plasmid and then the non-transduced cells (without reporter system incorporated) were selected via FACS and further somehow activated by molecules from the Prestwick library to express GFP which they can’t have.

2)      The author’s work is centered around the statement that CPX reprograms the gastric non-CSC into CSC cells. The experiments also show that CPX vastly decreases non-CSC cell viability. How can it be validated that the increase of CSC populations is due to the “true reprogramming”, aka conversion of previously non-CSC into CSC cells rather than relative selection/enrichment of CSC sub-population within the bulk gastric cancer cell pool due to death of non-CSC component after CPX treatment?

3)      It is important to show what the % of CSC-like cells within the wild-type AGS cell line is (without SORE reporter system insertion) and whether these cells exhibit phenotypic plasticity and retain CSC/non-CSC ratio over time. This has been reported for other cancer cell types. E.g., in ovarian cancer cells, FACS-mediated elimination of CSC from the ovarian cancer cell pool also generates a pure non-CSC population which over time re-gains its original CSC % within 3-5 days. If this is not the case for AGS cells, it is important to show that in a control experiment. Otherwise, the enrichment of CSC cells may not be due to the small molecule treatment but rather be a simple of CSC regain from sorted negative population over time.

4)      Paragraph 3.3, Figure 3: the authors show an increase in beta-gal activity in the cells post-CPX treatment and claim a linkage of senescence to CSC reprogramming towards stemness. Did they check the cells with enhanced beta-gal activity and the cells with SOX2 expression are the same cell sup-populations within the AGS cell bulk?

5)      Minor: line 406 – remove “To…” (clerical error)

Reviewer 2 Report

Pádua and co-authors prospected "small molecules" for converting gastric cancer cells into cancer stem-like cells (CSCs). The work provided interesting findings on CPX and the genesis of CSCs in gastric cancers. The main setback is the limited characterization of CSCs after CPX treatment. Further, authors should also prospect other CSCs markers (e.g., EMT-TFs) to better chacterize the resulting CPX-derived CSCs. 

Minor points

Introduction

Lines 68-79: this section should be revised. It is too focused on iPSC reprogramming, which is a different reprogramming scenario. Instead, authos should focus on the potential of small molecules to selectiveçy generate/maintain or kill CSCs (e.g., salinomycin), which have enormous scientific potential and may elucidate the long-lasting effects of cancer therapeutics;

Material and methods

Lines 101-102: Most cell lines were cultured in RPMI...;

Lines 123-124: Describe these methods in detail;

Line 152: replace rpm with rcf;

Lines 192-193: describe PCR cycling conditions;

Line 197: 2-ΔΔCT method;

Line 244: Cell proliferation assay;

Results

Line 266: cells are GFP negative, contrary...;

Line 283-284: avoid unecessary repetitive text (here and throughout the text), such as the multiple explanations of the GFP reporter; 

Discussion

Line 508: remove "bulk";

Line 510: therapy resistance and...;

Conclusions

Lines 582-585: This last sentence could be a cautionary note on the therapeutic potential of CPX since it may kill most tumor cells but may convert surviving cells into CSCs. 

Reviewer 3 Report

In this manuscript, the authors conducted a drug screen in gastric cancer cells and found that ciclopirox olamine (CPX) could reprogram cells to a cancer stem cell (CSC)-like phenotype. This is an interesting topic to study with potential clinical relevance. However, several concerns need to be addressed before publication.

1. How is the duration of treatment or dosages of CPX used in this study compared to that in the clinical setting? Here the effects were observed after 48h of treatment. But patients would likely undergo chemotherapy for a longer period. Will the reprogramming effect of CPX still occur with prolonged treatment in these cells? If not, then these results could have little relevance in the clinical setting.

2. Figure 1F: Is the percentage of GFP-positive cells calculated from viable cells or total cells?

3. How heterogeneous are AGS and Kato III cells? Are the cells that became GFP+ after CPX treatment intrinsically different from the GFP- ones that allowed them to turn on SOX2 expression?

4. Figure 2C: Is the decrease in %GFP+ cells due to loss of GFP expression or cell death? What is the cell viability/proliferation rate over 5 weeks after CPX treatment?

5. Figure 2D: Is the lower band in the SOX2 blot the specific band? If so, at 2 weeks the SOX2 level looks similar between 24h and 48h treatment.

6. Figure 2G: The c-MYC blot in AGS cells is pretty weak. Longer exposure or loading more lysate would be helpful. The c-MYC blot in Kato III cells is cut off at the bottom and needs to be replaced.

7. Figure 2I: AGS cells grow fewer but bigger colonies, while Kato III cells grow more but smaller ones. Why do they grow differently? Does it have anything to do with the higher levels of c-MYC, HIF1a, KLF4, etc., shown in Figure 5? Could this explain why Kato III is more resistant to CPX?

8. Figure 3A and 3B: Were the Seahorse experiment done using all the viable cells or just the GFP+ ones? If both GFP+ and GFP- cells were used, it would be hard to argue that the decrease in OXPHOS and increase in glycolysis is due to the reprogramming to CSCs rather than direct inhibition of OXPHOS by CPX.

9. The percentage of senescent cells is much higher than that of GFP+ cells. Is there any overlap between these two populations? Are the GFP+ cells also senescent?

10. Figure 4A: What are the changes in cell number over time? Is it possible that senescent cells died off and detached from the plate over time, which leads to the decrease observed in their percentage?

11. Figure 4B-4D: Are the GFP+ cells after the first CPX treatment more resistant to a second exposure? Do they proliferate faster? Do the GFP+ cells also have higher BrdU incorporation? Which contributes to the CSC reprogramming and CPX resistance more, SOX2 and/or OCT4 expression or induction of senescence?

12. Figure 5G: What is the level of SOX2 in SORE6- vs SORE6+ AGS or Kato III cells?

13. Kato III cells seem to have higher basal levels of SOX2, MYC, etc compared to AGS cells. Are they more aggressive? Do they grow faster or adopt more of the metabolic features like the CPX-treated cells?

Round 2

Reviewer 3 Report

The authors have addressed the comments except for Figure 2G, where the western blots for c-MYC didn't seem to be replaced/updated in the revised manuscript or in the original images for blots. 
